# Optimal Decisions in Green, Low-Carbon Supply Chain Considering the Competition and Cooperation Relationships between Different Types of Manufacturers

**DOI:** 10.3390/ijerph192215111

**Published:** 2022-11-16

**Authors:** Xiaoqing Zhang, Wantong Chen, Min Wang, Dalin Zhang

**Affiliations:** 1Business School, Jiangsu Normal University, Xuzhou 221116, China; 2School of Business, Linyi University, Linyi 276000, China; 3Department of Computer Science, Aalborg University, 9220 Aalborg, Denmark

**Keywords:** green low-carbon supply chain, recycling and remanufacturing, level of recycling effort, products’ greenness level

## Abstract

In this study, we built a green, low-carbon supply chain including one green manufacturer, one green remanufacturer and one retailer in which the manufacturer produces new, green, low-carbon products and the remanufacturer recycles and remanufactures the green, low-carbon products. We assumed the manufacturer to be the Stackelberg leader and the remanufacturer and the retailer to be Stackelberg followers. The game model was solved using backward induction. We discuss the optimal operation strategies for green, low-carbon supply-chain members in a centralized decision-making model, decentralized decision-making model, manufacturer–remanufacturer cooperative decision-making model and manufacturer–retailer cooperative decision-making model. Furthermore, we discuss the impacts of the unit cost savings for remanufacturing, the recovery cost coefficient and the green improvement cost coefficient on the green supply-chain members’ optimal decision and profits. The results show that increased unit cost savings from remanufacturing can increase the total profit of the supply chain and promote the recycling and remanufacturing of waste products. Moreover, the total profit of the green, low-carbon supply chain is the highest in the centralized decision-making model and lowest in the manufacturer cooperative decision-making model. When there is a cooperation relationship between the manufacturer and the retailer, the optimal recycling effort level and the optimal greenness level for the new product and the remanufactured product are the highest.

## 1. Introduction

### 1.1. Background and Research Motivation

With the rapid development of the social economy, such problems as environmental pollution, resource waste and ecological destruction are being taken increasingly seriously, and the environmental-protection consciousness of customers has gradually grown. In addition, many countries and regions have introduced various strict environmental protection regulations. For example, the Chinese government advocates new energy and green development in the 13th Five-Year National Development Plan (Swami and Shah [1] and Chen [2]). In recent years, with the increasing awareness of environmental protection among consumers and in order to maintain long-term competitive advantages in the market, many firms in the supply chain have also begun to implement green supply-chain management. A green supply chain is defined as a modern supply-chain management mode that considers environmental impact and resource efficiency in the supply chain. The purpose is to make products across the whole process, from material acquisition, processing, manufacturing, packaging, storage, transportation and use to scrap and the recycling process. This supply chain has the lowest impact on the environment and the most efficient use of resources. In practice, many firms have reaped benefits from green supply-chain management. For example, through the implementation of green supply-chain management, Beijing Automotive Co., Ltd. has ensured that the design and development, production and manufacturing, use and maintenance and recycling of its vehicle products all meet environmental protection laws and standards.

At the same time, in recent years, global warming has been further aggravated and has gradually become a hot spot for the international community. The development of a low-carbon economy, energy conservation and emission reduction have become some of the major strategic measures used to promote the sustainable development of the global economy. Therefore, many countries around the world have introduced relevant carbon emission reduction plans. For example, the Chinese government proposed to reduce the carbon dioxide emission intensity per unit of GDP by 40% in 2020 compared to 2005. As one important way to achieve carbon emission reduction, a carbon emission tax has been implemented in some countries, such as Sweden, Canada, Australia and so on. Thus, while producing green products, firms should also actively adopt various technologies and methods to reduce carbon emissions. To realize the win-win situation of enterprise development and environmental protection, it is necessary for many more firms to introduce green, low-carbon supply-chain management in business management practice. As is known, the design and production of green, low-carbon products is an important link in green, low-carbon supply-chain management, and it can contribute significant improvements to customer satisfaction, market share and profits. Thus, for firms, to make the appropriate selling channel decisions, the issue of green, low-carbon products is worth thinking deeply about.

In addition, the recycling and reuse of used green, low-carbon products have very important theoretical and practical significance. The recycling of used green, low-carbon products can enhance environmental protection and help achieve sustainable economic and social development. The utilization of resource circulation and the development of green energy are effective ways to address resource shortages and protect the environment. Perfecting the recycling system for waste products and improving the level of utilization of renewable resources are beneficial for protecting the environment and alleviating the shortage of resources problem. Furthermore, the recycling of used green, low-carbon products by enterprises can help effectively improve the efficiency of resource utilization. Of course, the recycling of green, low-carbon products can not only help enterprises achieve environmental targets but can also reduce manufacturing costs and improve enterprises’ brand image. Thus, for recycling and remanufacturing firms, how to develop appropriate collection strategies and choose the appropriate collection channel and the appropriate cooperation strategy are also issues worth paying attention to.

Motivated by the above analysis, this study attempted to answer the following research questions:(1)What is the optimal operating strategy in green, low-carbon supply chains under different decision-making models?(2)What are the impacts of remanufacturing unit cost savings, the recovery cost coefficient and the green input cost coefficient on the optimal decision and enterprises’ profit?

Large numbers of scholars have discussed the selling channel strategy in forward supply chains and the collection channel strategy in traditional green supply chains. Some researchers have discussed operation management in green supply-chain management. However, few studies have analyzed the competition and cooperation problem in green, low-carbon supply chains. Filling this gap, this is the first time a study has addressed the green, low-carbon supply chain as a research object. Moreover, we introduce the competition and cooperation situation into the green, low-carbon supply chain.

Thus, we considered a green, low-carbon supply chain consisting of one green manufacturer, one green remanufacturer and one retailer. We built four different decision-making models, including a centralized decision-making model, a decentralized decision-making model, the cooperative decision-making model involving the manufacturer and the remanufacturer and the cooperative decision-making model involving the manufacturer and the retailer. We mainly discuss the level of recycling effort for waste green, low-carbon products, the degree of the green design of green, low-carbon products and the pricing decision problems. This paper provides meaningful theoretical guidance for the recycling and remanufacturing of the green, low-carbon products.

The following results were obtained. First, an increase in the unit cost savings for remanufacturing can make a firm or alliance engaged in remanufacturing in a green, low-carbon supply chain pay much more attention to recycling efforts and produce more remanufactured green, low-carbon products. Second, with the increase in the recycling cost coefficient and the green investment cost coefficient, the optimal recycling effort level and the optimal greenness level for the remanufactured product decrease, and the total profit of the green, low-carbon supply chain decreases. Last, when there is a cooperation relationship between the manufacturer and the retailer, the optimal recycling effort level and the optimal greenness level of the new low-carbon product and the remanufactured low-carbon product are the highest.

### 1.2. Contribution Statement and Paper Structure

This study contributes to the literature by considering the optimal decisions for the green, low-carbon supply chain and different types of manufacturers. This study brings together studies on the selling channel selection strategy and the collection channel selection strategy in green supply chains. The impacts of the remanufacturing cost savings, the recovery cost coefficient and the green input cost coefficient on the optimal decisions and firms’ profits are discussed. 

The remainder of this paper is organized as follows. Section 2 provides the literature review. Section 3 builds the model. The equilibrium analysis is provided in Section 4. The results analysis is provided in Section 5. Finally, the conclusions are discussed in Section 6.

## 2. Literature Review

At present, the literature related to this paper is mainly concentrated on the following two aspects: (1) the selling channel selection strategy in green supply chains; (2) the collection channel selection strategy in green supply chains. 

### 2.1. The Selling Channel Selection Strategy in Green Supply Chains

Recently, many scholars have begun to notice the green supply chain. For example, Swami and Shah [1] analyzed the selling channel selection problem in green supply-chain management (GSCM) considering green investment, and the results showed that green investment has a very important effect on selling channel selection. Chen [2] analyzed the effect of the government’s environmental standards on manufacturers’ selling channel selection in GSCM. Hsu et al. [3] analyzed the impact of customers’ environmental protection awareness on selling channel selection. They pointed out that environmental awareness could improve a firm’s profit in GSCM. Wang et al. [4] studied the effect of government subsidies on members’ revenues in GSCM, and the results showed that they have a positive influence on the selling channel selection. Zhang and Li [5] discussed the optimal results for a point firm in GSCM and pointed out that the greenness level has a vast impact on the GSCM. Dey et al. [6] and Yan et al. [7] discussed the effects of different pricing contracts on greenness level decisions in two-period GSCM.

Khorshidvand et al. [8] analyzed the impacts of advertisement levels and green policy on optimal decisions in GSCM. Qu et al. [9] analyzed optimal pricing decisions in GSCM using a three-level multi-criteria decision method. Gong et al. [10] analyzed five different situations’ cost constraint models in GSCM and pointed out that individual preference could influence the optimal selection. Bhatia et al. [11] reviewed status of research on green supply-chain management by searching through the relevant literature. Ofek et al. [12] and Wang et al. [13] analyzed the effects of a product’s difference and the channel operation cost on a retailer’s channel selection in GSCM. The above studies discussed the selling channel strategy problem in traditional green supply chains. However, they did not analyze collection channel competition and cooperation problems in green supply chains. Thus, in our paper, we discuss the competition and cooperation problem in GSCM using game theory.

### 2.2. The Collection Channel Selection Strategy in Green Supply Chains

Nowadays, with the continuous development of information technology, the collection and reuse of used green products has become a very important problem. Green, low-carbon supply-chain management is one of the most important ways to realize the recovery and reuse of the old green products. Many scholars have discussed this problem. For example, Savaskan et al. [14] discussed the optimal collection channel selection strategy in traditional supply chains and pointed out that the third-party collection strategy was a good choice for the manufacturer. Mukhopadhyay et al. [15] analyzed the impact of the customer’s preference on the retailer’s collection strategy. Guide [16] analyzed the effect of the product’s value attenuation on the collection effect and pointed out that there is a positive relationship between the value attenuation and the firm’s collection effect. Webster et al. [17] and Hong and Yeh [18] pointed out that the third-party collection style is much more popular in green supply chains. Peral [19] analyzed the coordination of an integrated green supply chain including one manufacturer and one retailer. Chuang et al. [20] analyzed the collection channel selection problem in GSCM among high-tech firms. The above papers mainly analyzed the collection channel selection problem in GSCM. However, they did not analyze the collection channel competition and cooperation problem in GSCM.

Many scholars have started to analyze and discuss the coordination problem in the collection channel in GSCM. For example, Giovanni [21] analyzed the advantages and disadvantages of different collection channels used by manufacturers and pointed out that manufacturers can use a third-party collection channel. He and Xu [22] pointed out that new collection technology could have important impacts on the collection of used green products. Ayvaz et al. [23] designed a popular reverse logistics network and analyzed the optimal reverse channel selection problem under this model. Giovanni [24] pointed out that both the manufacturers and retailers could use different collection strategies when they recycle used green products. Yi et al. [25] and Polat et al. [26] analyzed the optimal pricing and collection strategy in a single and multiple period(s).

In short, large numbers of scholars have discussed the selling channel strategy in forward supply chains and the collection channel strategy in traditional green supply chains. Some studies have analyzed operation management in green supply-chain management. However, the above studies do not analyze the impacts of remanufacturing cost savings, the recovery cost coefficient and the green input cost coefficient on optimal decisions and firms’ profit. Moreover, no studies have discussed the competition and cooperation problem in green, low-carbon supply chains. Filling this gap, this is the first time that the green, low-carbon supply chain has been taken as a research object. Moreover, we introduce the competition and cooperation situation into the green, low-carbon supply chain and analyze the competition and cooperation problem involving the green manufacturer, green remanufacturer and retailer by using a game model. Table 1 shows the difference between our paper and previous research.

## 3. The Model

### 3.1. Model Setup

In this study, we built a green, low-carbon supply chain consisting of one manufacturer, one retailer and one remanufacturer in which the green manufacturer (*Mn*) is responsible for the production of the new green, low-carbon products and the green remanufacturer (*Mr*) recycles the green, low-carbon products. Then, the retailer (R) sells the new and remanufactured green, low-carbon products to customers. We assumed that all firms hold sufficient inventory to meet the demand (see Tang et al. [27]). Thus, we built four different green, low-carbon supply chain theoretical models (Figure 1).

(i).The centralized decision-making model (model C): In this model, the manufacturer, the remanufacturer and the retailer constitute an alliance. The decision order is that the alliance first decides on the level of recycling effort e and then the greenness levels $g_{n}$ and $g_{r}$ and the retail prices $p_{n}$ and $p_{r}$ of the new products and the remanufactured products;(ii).The decentralized decision-making model (model D): In this model, the manufacturer and the remanufacturer are considered the Stackelberg game leaders and make decisions simultaneously, while the retailer is the Stackelberg game follower. The decision order is that the green remanufacturer decides on the level of recycling effort e before the selling season and then the manufacturer and the remanufacturer decide on the greenness levels $g_{n}$ and $g_{r}$. Last, the retailer decides on the retail prices $p_{n}$ and $p_{r}$ of the new products and the remanufactured products;(iii).The cooperative decision-making model involving the manufacturer and the remanufacturer (model M-M): In this model, the manufacturer and the remanufacturer constitute an alliance. The decision order is that the alliance decides on the level of recycling effort e and then they decide on the greenness levels $g_{n}$ and $g_{r}$. Last, the retailer decides on the retail prices $p_{n}$ and $p_{r}$ of the new products and the remanufactured products;(iv).The cooperative decision-making model involving the manufacturer, the retailer and the remanufacturer (model R-M-M): The manufacturer and the retailer here constitute an alliance (*RMn*) and, at the same time, the remanufacturer and the retailer constitute an alliance (*RMr*). The decision order is that the alliance *RMr* decides on the level of recycling effort e and then they decide on the greenness levels $g_{n}$ and $g_{r}$ and the retail prices $p_{n}$ and $p_{r}$ of the new products and the remanufactured products.

### 3.2. Demand and Parameters

In this study, we focused on the competition between greenness levels of the new products and the remanufactured products in order to highlight this core research question; we did not consider the price competition between the new products and the remanufactured products. Thus, we assumed that the market demand was related to the retail price of the product and its greenness level, to the greenness level of the other products and to the carbon emissions per unit of the product. As a result, the demand function of the new products can be expressed as follows:(1)$$d_{n}=\alpha_{n}-p_{n}+g_{n}-\beta g_{r}-\tau f$$

The demand function of the remanufactured products can expressed as follows:(2)$$d_{r}=\alpha_{r}-p_{r}+g_{r}-\beta g_{n}-\tau f$$

The same demand functional form is used in the existing literature. To simplify the model, we assumed that the potential market sizes for the new and the remanufactured products were the same ($\alpha_{n}=\alpha_{r}=\alpha$).

The level of recycling effort of the green remanufacturer (*Mr*) is e and the recycling cost is $\theta e^{2}$, where θ is the coefficient of the recycling cost. The green manufacturer *Mn*’s green input cost is $\lambda g_{n}^{2}$ and the green remanufacturer *Mr*’s green input cost is $\lambda g_{r}^{2}$, where λ is the coefficient of the green input cost. The unit production cost of the new green products is $c_{n}$ and the unit production cost of the new remanufactured green products is $c_{r}$. The wholesale price of the new green products is $w_{n}$ and the wholesale price of the new remanufactured green products is $w_{r}$. Furthermore, we assumed that $c_{n}>c_{r}$ and $w_{n}>w_{r}$. Thus, the unit cost savings of the remanufacturer’s green product are $\delta =c_{n}-c_{r}>0$, and the average unit production cost of the remanufactured products is ${\bar{c}}_{r}=c_{n}\left(1-e\right)+c_{r}e=c_{n}-\delta e$. All parameters are listed in Table 2. 

In the rest of this paper, the subscript n represents the new green products, the subscript r represents the remanufactured green products, the subscript Mn represents the green manufacturer, the subscript Mr denotes the green remanufacturer, the subscript R denotes the retailer, the subscript M denotes the alliance between the manufacturer and the remanufacturer, the subscript RMn denotes the alliance between the manufacturer and the retailer and the subscript RMr denotes the alliance between remanfacturer and the retailer. The superscript notations C,D,M−M,R−M−M denote the four different decision-making models.

## 4. Equilibrium Analysis

In this section, the above four green, low-carbon supply chain theoretical models are respectively applied, and we obtain the optimal decision for each supply chain member.

### 4.1. The Centralized Model (Model C)

In this model, the manufacturer, the remanufacturer and the retailer constitute an alliance. The decision order is that the alliance decides on the level of recycling effort e and then the greenness levels $g_{n}$ and $g_{r}$ and the retail prices $p_{n}$ and $p_{r}$ of the new products and the remanufactured products. The decision problem is as follows:(3)$$Max\begin{array}{cc} & \pi^{C} \end{array}=\left(p_{n}-c_{n}\right)d_{n}+\left(p_{r}-c_{n}+\delta e\right)d_{r}-\frac{1}{2}\lambda g_{n}^{2}-\frac{1}{2}\lambda g_{r}^{2}-\frac{1}{2}\theta e^{2}-\eta {\left(f_{0}-f\right)}^{2}$$

We can solve the above equation by using the backward induction method and obtain Proposition 1.

**Proposition** **1**.
*The equilibrium results are derived as follows:*
(i).
*The equilibrium level of the recycling effort is as follows:*

$$e^{C*}=\frac{\delta \left(\alpha -c_{n}\right)\left[{\left(\beta^{2}+1\right)}^{2}-\lambda \right]+f}{4\lambda \theta \left(\beta^{2}+1-\lambda \right)-\beta^{2}\theta \left(\beta^{2}-2\right)-\theta -\left(A_{1}+2\right)\lambda \delta^{2}};$$

(ii).
*The equilibrium greenness levels are as follows:*

$$g_{n}^{C*}=\frac{\left(1-\beta \right)\left(\alpha -c_{n}\right)\left[{\left(\beta +1\right)}^{2}-\lambda \right]-X_{1}\beta \delta +f}{\beta^{2}\left(2X_{1}-\beta^{2}\right)+{\left(2\lambda +1\right)}^{2}},\;\mathrm{and}\;g_{r}^{C*}=\frac{\left[\beta \left(X_{1}+\beta \right)+2\lambda -1\right]\left(\alpha -c_{n}\right)+\beta \left(X_{1}+\beta -1\right)\delta +f}{\beta^{2}\left(2X_{1}-\beta^{2}\right)+{\left(2\lambda +1\right)}^{2}};$$

(iii).*The equilibrium retail prices are as follows:*$$p_{n}^{C*}=\frac{\alpha X_{2}+c_{n}X_{3}-2\lambda \beta \delta }{\beta^{2}\left(2X_{1}-\beta^{2}\right)+{\left(2\lambda +1\right)}^{2}},\;\mathrm{and}\;p_{r}^{C*}=\frac{\alpha X_{2}+c_{n}X_{3}+\left(2\lambda \beta -X_{3}-3\lambda \right)\delta }{\beta^{2}\left(2X_{1}-\beta^{2}\right)+{\left(2\lambda +1\right)}^{2}}.$$*where* $X_{1}=2\beta^{2}-\lambda +1$, $X_{2}=2\lambda \left(\lambda +1-\beta^{2}+\beta \right)$*and*$X_{3}=\beta^{2}+\left(\lambda +1\right)\beta^{2}+\lambda \beta$.


### 4.2. The Decentralized Model (Model D)

In this model, the manufacturer and the remanufacturer are considered the Stackelberg game leaders and make decisions simultaneously, while the retailer is the Stackelberg game follower. The decision order is that the green remanufacturer decides on the level of recycling effort e before the selling season and then the manufacturer and the remanufacturer decide on the greenness levels $g_{n}$ and $g_{r}$. Last, the retailer decides on the retail prices $p_{n}$ and $p_{r}$ of the new products and the remanufactured products. The decision problem is as follows:(4)$$Max\begin{array}{cc} & \pi_{Mn}^{D} \end{array}=\left(w_{n}-c_{n}\right)d_{n}-\frac{1}{2}\lambda g_{n}^{2}-\eta \left(f_{0}-f\right)$$
(5)$$Max\begin{array}{cc} & \pi_{Mr}^{D} \end{array}=\left(w_{r}-c_{n}+\delta e\right)d_{r}-\frac{1}{2}\lambda g_{r}^{2}-\frac{1}{2}\theta e^{2}-\eta \left(f_{0}-f\right)$$
(6)$$Max\begin{array}{cc} & \pi_{R}^{D}=\left(p_{n}-w_{n}\right) \end{array}d_{n}+\left(p_{r}-w_{r}\right)d_{r}$$

We can solve the above equations by using the backward induction method and obtain Proposition 2.

**Proposition** **2**.
*The equilibrium results are derived as follows:*
(i).
*The equilibrium level of the recycling effort is as follows:*

$$e^{D*}=\frac{\left[\lambda \left(\alpha -w_{r}\right)+\beta \left(c_{n}-w_{n}\right)+w_{r}+c_{n}\right]\delta +f}{2\lambda \theta +\delta^{2}};$$

(ii).
*The equilibrium greenness levels are as follows:*

$$g_{n}^{D*}=\frac{w_{n}+c_{n}+f}{2\lambda }\mathrm{and}\;g_{r}^{D*}=\frac{w_{r}+c_{n}+\delta +f}{2\lambda };$$

(iii).
*The equilibrium retail prices are as follows:*

$$p_{n}^{D*}=\frac{\alpha +g_{n}^{D*}-\beta g_{r}^{D*}+w_{n}}{2}\;\mathrm{and}\;p_{r}^{D*}=\frac{\alpha +g_{r}^{D*}-\beta g_{n}^{D*}+w_{r}}{2}$$




### 4.3. The Cooperative Model Involving the Manufacturer and the Remanufacturer (Model M-M)

In this model, the manufacturer and the remanufacturer constitute an alliance. The decision order is that the alliance decides on the level of recycling effort e and then they decide on the greenness levels $g_{n}$ and $g_{r}$. Last, the retailer decides on the retail prices $p_{n}$ and $p_{r}$ of the new products and the remanufactured products. The decision problem is as follows:(7)$$Max\begin{array}{cc} & \pi_{M}^{M-M}=\left(w_{n}-c_{n}\right) \end{array}d_{n}+\left(w_{r}-c_{n}+\delta e\right)d_{r}-\frac{1}{2}\lambda g_{n}^{2}-\frac{1}{2}\lambda g_{r}^{2}-\frac{1}{2}\theta e^{2}-\eta \left(f_{0}-f\right)$$
(8)$$Max\begin{array}{cc} & \pi_{R}^{M-M} \end{array}=\left(p_{n}-w_{n}\right)d_{n}+\left(p_{r}-w_{r}\right)d_{r}$$

We can solve the above equations by using the backward induction method and obtain Proposition 3.

**Proposition** **3**.
*The equilibrium results are derived as follows:*


(i).    *The equilibrium level of the recycling effort is as follows:*$$e^{M-M*}=\frac{\left(\beta^{2}+1\right)\left(w_{r}-c_{n}\right)+2\lambda \left(\alpha -w_{r}\right)+2\beta \left(c_{n}+w_{n}\right)+f}{2\lambda \theta +\delta^{2}\left(\beta^{2}+1\right)}$$

(ii).    *The equilibrium greenness levels are as follows:*$$g_{n}^{M-M*}=\frac{\beta \left(c_{n}-w_{r}\right)+w_{n}-c_{n}+\beta \delta +f}{2\lambda }\;\mathrm{and}\;g_{r}^{M-M*}=\frac{\beta \left(c_{n}-w_{n}\right)+w_{r}-c_{n}+\delta +f}{2\lambda };$$

(iii).   *The equilibrium retail prices are as follows:*$$p_{n}^{M-M*}=\frac{\alpha +g_{n}^{M-M*}-\beta g_{r}^{M-M*}+w_{n}}{2}\;\mathrm{and}\;p_{r}^{M-M*}=\frac{\alpha +g_{r}^{M-M*}-\beta g_{n}^{M-M*}+w_{r}}{2}.$$

### 4.4. The Cooperative Model Involving the Manufacturer, the Retailer and the Remanufacturer (Model R-M-M)

In this model, the cooperative decision-making model involving the manufacturer, the retailer and the remanufacturer (model *R-M-M*), the manufacturer and the retailer constitute an alliance (*RMn*); at the same time, the remanufacturer and the retailer constitute an alliance (*RMr)*. The decision order is that the alliance *RMr* decide on the level of recycling effort e and then they decide on the greenness levels $g_{n}$ and $g_{r}$ and the retail prices $p_{n}$ and $p_{r}$ of the new products and the remanufactured products. The decision problem is as follows:(9)$$Max\begin{array}{cc} & \pi_{RMn}^{R-M-M} \end{array}=\left(p_{n}-c_{n}\right)d_{n}-\frac{1}{2}\lambda g_{n}^{2}-\eta \left(f_{0}-f\right)$$
(10)$$Max\begin{array}{cc} & \pi_{RMr}^{R-M-M} \end{array}=\left(p_{r}-c_{n}+\delta e\right)d_{r}-\frac{1}{2}\lambda g_{r}^{2}-\frac{1}{2}\theta e^{2}-\eta \left(f_{0}-f\right)$$

We can solve the above equations by using the backward induction method and obtain Proposition 4.

**Proposition** **4**.
*The equilibrium results are derived as follows:*
(i).
*The equilibrium level of the recycling effort is as follows:*

$$e^{R-M-M*}=\frac{\delta \lambda {\left(2\lambda -1\right)}^{2}\left(\alpha -c_{n}\right)X_{4}+f}{\left(\delta^{2}\lambda \left(2\lambda^{3}-2\lambda^{2}+2\lambda -1\right)+\beta^{2}\theta \left(2\lambda^{2}-2\lambda -\beta^{2}+2\right)+2\lambda \theta \left(1-2\lambda^{3}+2\lambda^{2}-2\lambda \right)-\theta \right)}$$

(ii).
*The equilibrium greenness levels are as follows:*

$$g_{n}^{R-M-M*}=\frac{\left(\alpha -c_{n}\right)X_{4}+\beta \delta +f}{\beta^{2}-{\left(2\lambda -1\right)}^{2}}\;\mathrm{and}\;g_{r}^{R-M-M*}=\frac{\left(\alpha -c_{n}\right)X_{4}+\left(1-2\lambda \right)\delta +f}{\beta^{2}-{\left(2\lambda -1\right)}^{2}}.$$

(iii).*The equilibrium retail prices are as follows:*$$p_{n}^{M-M*}=\frac{\left(\beta -\lambda \right)\alpha +\left(X_{1}+7\lambda -\beta \right)c_{n}+\beta \lambda \delta }{\beta^{2}-{\left(2\lambda -1\right)}^{2}}\;\mathrm{and}\;p_{r}^{R-M-M*}=\frac{\alpha \lambda X_{4}+\left(X_{1}-\beta \lambda -2\lambda^{2}+5\lambda \right)c_{n}+\left(2\lambda^{2}-5\lambda -X_{1}\right)\delta }{\beta^{2}-{\left(2\lambda -1\right)}^{2}}$$*where* $X_{4}=\beta -2\lambda +1$.


## 5. Result Analysis

In this section, we analyze the impacts of the unit cost savings for remanufacturing δ, the coefficient of the recycling cost θ and the coefficient of the green input cost λ on the equilibrium decisions and the firms’ profits.

### 5.1. The Impact of the Unit Cost Savings for the Remanufacturing

By comparing the above equilibrium solutions, we can deduce the following lemmas.

**Lemma** **1**.
*In model C, the impact of *

δ

*on the equilibrium solutions is as follows:*



$$(1)\;\frac{\partial e^{C*}}{\partial \delta }>0;\;(2)\;\frac{\partial g_{n}^{C*}}{\partial \delta }<0;\;(3)\;\frac{\partial g_{r}^{C*}}{\partial \delta }>0;\;(4)\;\frac{\partial p_{n}^{C*}}{\partial \delta }<0;\;(5)\;\frac{\partial p_{r}^{C*}}{\partial \delta }<0.$$


From Lemma 1, we deduce that, in model C, the optimal level of recycling effort $e^{C*}$ and the optimal greenness level of the remanufactured products $g_{r}^{C*}$ increase with the increases in δ. However, the optimal greenness level of the new products $g_{n}^{C*}$ decreases. This suggests that the unit cost savings of remanufacturing δ are good for the remanufactured products but bad for the new products. In addition, with the increases in δ, the retail prices of the two kinds of products decrease. The reason is that the whole supply chain can save much more in production costs from the remanufacturing process, which reduces the retail price of the green products and allows a much greater market share to be obtained. 

**Lemma** **2**.
*In model D, the impact of *

δ

*on the equilibrium solutions is as follows:*




$$(1)\frac{\partial e^{D*}}{\partial \delta }>0;\;(2)\;\frac{\partial g_{r}^{D*}}{\partial \delta }>0;\;(3)\frac{\partial p_{n}^{D*}}{\partial \delta }<0;\;(4)\frac{\partial p_{r}^{D*}}{\partial \delta }>0.$$



From Lemma 2, we deduce that, in model D, δ has positive impacts on the optimal level of recycling effort $e^{D*}$, the optimal greenness level of the remanufactured products $g_{r}^{D*}$ and the optimal retail price $p_{r}^{D*}$. This suggests that the unit cost savings of remanufacturing are good for the production of the remanufactured products. In addition, with the increases in δ, the optimal retail price of the new green products $p_{n}^{D*}$ decreases. This is because the improvement in the greenness level of remanufactured products is not conducive to the sale of new products. In this case, the retailer can only increase the sale quantity by reducing the retail price of new products.

**Lemma** **3**.
*In model M-M, the impact of *

δ

*on the equilibrium solutions is as follows:*




$$(1)\;\frac{\partial e^{M-M*}}{\partial \delta }>0;\;(2)\;\frac{\partial g_{n}^{M-M*}}{\partial \delta }<0;\;(3)\frac{\partial g_{r}^{M-M*}}{\partial \delta }>0;\;(4)\frac{\partial p_{n}^{M-M*}}{\partial \delta }<0;\;(5)\frac{\partial p_{r}^{M-M*}}{\partial \delta }>0.$$



From Lemma 3, we deduce that, in model M-M, δ has positive impacts on the optimal level of recycling effort $e^{M-M*}$, the optimal greenness level of the remanufactured products $g_{r}^{M-M*}$ and the optimal retail price $p_{r}^{M-M*}$. This suggests that the competitive relationship between the manufacturer and the remanufacturer does not impact the remanufacturing unit cost savings from remanufacturing activities. At the same time, with the increases in δ, the optimal greenness level of the new products $g_{n}^{M-M*}$ decreases. This also suggests that, in the case of the cooperation between the manufacturer and the remanufacturer, when the remanufactured product is much more advantageous, the manufacturer alliance will choose to reduce the greenness level of the new product so as to reduce the green production cost. 

**Lemma** **4**.
*In model R-M-M, the impact of *

δ

*on the equilibrium solutions is as follows:*




$$(1)\;\frac{\partial e^{R-M-M*}}{\partial \delta }>0;\;(2)\;\frac{\partial g_{n}^{R-M-M*}}{\partial \delta }<0;\;(3)\frac{\partial g_{r}^{R-M-M*}}{\partial \delta }>0;\;(4)\frac{\partial p_{n}^{R-M-M*}}{\partial \delta }<0;\;(5)\frac{\partial p_{r}^{R-M-M*}}{\partial \delta }<0.$$



Lemma 4 is similar to Lemma 1. This is because, in model C and in model R-M-M, the degree of cooperation between the manufacturer and the retailer is relatively high, and the unit cost savings advantage from remanufacturing can be shared by the supply chain members.

### 5.2. The Impact of the Coefficient of the Recycling Cost

By comparing the above equilibrium solutions, we can deduce following lemmas.

**Lemma** **5**.
*In model C, the impact of *

θ

*on the equilibrium solutions is as follows:*




$$(1)\;\frac{\partial e^{C*}}{\partial \theta }<0;\;(2)\;\frac{\partial g_{n}^{C*}}{\partial \theta }>0;\;(3)\frac{\partial g_{r}^{C*}}{\partial \theta }<0;\;(4)\frac{\partial p_{n}^{C*}}{\partial \theta }>0;\;(5)\frac{\partial p_{r}^{C*}}{\partial \theta }>0.$$



From Lemma 5, we deduce that, in model C, the optimal level of recycling effort $e^{C*}$ and the optimal greenness level of the remanufactured products $g_{r}^{C*}$ decrease with the increases in θ. However, the optimal greenness level of the new products $g_{n}^{C*}$ increases. This suggests that the increase in the recovery cost is not conducive to the production of remanufactured products, but it is beneficial to the production of new products. In addition, with the increases in θ, the retail prices of two kinds of products increase. The reason is that the firm must use more revenue form product sales to offset the increased recovery cost.

**Lemma** **6**.
*In model D, the impact of *

θ

*on the equilibrium solutions is as follows:*




$$(1)\;\frac{\partial e^{D*}}{\partial \theta }<0;\;(2)\;\frac{\partial g_{r}^{D*}}{\partial \theta }<0;\;(3)\;\frac{\partial p_{n}^{D*}}{\partial \theta }>0;\;(4)\;\frac{\partial p_{r}^{D*}}{\partial \theta }<0.$$



From Lemma 6, we deduce that, in model D, θ has a negative impact on the optimal level of the recycling effort $e^{D*}$, the optimal greenness level of the remanufactured products $g_{r}^{D*}$ and the optimal retail price $p_{r}^{D*}$. In addition, with the increases in θ, the optimal retail price of the new green products $p_{n}^{D*}$ increases. This suggests that the increase in recovery cost is not conducive to the production of remanufactured products. At the same time, the new green products have more advantages in the market, and their optimal retail price increases accordingly.

**Lemma** **7**.
*In model M-M, the impact of *

θ

*on the equilibrium solutions is as follows:*




$$(1)\;\frac{\partial e^{M-M*}}{\partial \theta }<0;\;(2)\;\frac{\partial g_{n}^{M-M*}}{\partial \theta }>0;\;(3)\;\frac{\partial g_{r}^{M-M*}}{\partial \theta }<0;\;(4)\;\frac{\partial p_{n}^{M-M*}}{\partial \theta }>0;\;(5)\;\frac{\partial p_{r}^{M-M*}}{\partial \theta }<0.$$



From Lemma 7, we deduce that, in model M-M, θ has a positive impact on the optimal greenness level of the new products $g_{n}^{M-M*}$. This is because when there is a cooperative relationship between the manufacturer and the remanufacturer, the recycling behavior of the remanufacturer will have a greater impact on their respective decisions.

**Lemma** **8**.
*In model R-M-M, the impact of *

θ

*on the equilibrium solutions is as follows:*




$$(1)\;\frac{\partial e^{R-M-M*}}{\partial \theta }<0;\;(2)\;\frac{\partial g_{n}^{R-M-M*}}{\partial \theta }>0;\;(3)\;\frac{\partial g_{r}^{R-M-M*}}{\partial \theta }<0;\;(4)\;\frac{\partial p_{n}^{R-M-M*}}{\partial \theta }>0;\;(5)\;\frac{\partial p_{r}^{R-M-M*}}{\partial \theta }>0.$$



Lemma 8 is similar to Lemma 5. The increase in recovery cost has a negative impact on the recycling of waste products and the production of remanufactured products and a positive impact on the production of new products.

Lemma 5 is similar to Lemma 1 This is because in models C and R-M-M, the degree of cooperation between the manufacturer and the retailer is relatively high, and the unit cost savings advantage from remanufacturing can be shared by the supply chain members.

### 5.3. The Impact of the Carbon Emission per Unit Product 

By comparing the above equilibrium solutions, we can deduce the following lemmas.

**Lemma** **9**.
*In model C, the impact of *

f

*on the equilibrium solutions is as follows:*




$$(1)\;\frac{\partial e^{C*}}{\partial f}>0;\;(2)\;\frac{\partial g_{n}^{C*}}{\partial f}>0;\;(3)\;\frac{\partial g_{r}^{C*}}{\partial f}>0.$$



**Lemma** **10**.
*In model D, the impact of *

f

*on the equilibrium solutions is as follows:*




$$(1)\;\frac{\partial e^{D*}}{\partial f};\;(2)\;\frac{\partial g_{r}^{D*}}{\partial f}>0;\;(3)\;\frac{\partial g_{n}^{D*}}{\partial f}>0$$



**Lemma** **11**.
*In model M-M, the impact of *

f

*on the equilibrium solutions is as follows:*




$$(1)\;\frac{\partial e^{M-M*}}{\partial f}>0;\;(2)\;\frac{\partial g_{n}^{M-M*}}{\partial f}>0;\;(3)\;\frac{\partial g_{r}^{M-M*}}{\partial f}>0.$$



**Lemma** **12**.
*In model R-M-M, the impact of *

f

*on the equilibrium solutions is as follows:*




$$(1)\;\frac{\partial e^{R-M-M*}}{\partial f}>0;\;(2)\;\frac{\partial g_{n}^{R-M-M*}}{\partial f}>0;\;(3)\;\frac{\partial g_{r}^{R-M-M*}}{\partial f}>0.$$



From all the above lemmas, we can deduce that the carbon emission per unit product in each theoretical model has a positive impact on the the level of recycling effort of the green remanufacturer e, the greenness level of the new products $g_{n}$ and the greenness level of the remanufactured products $g_{r}$. In other words, with the increase in the carbon emission per unit product, the greenness level of the new or the remanufactured products can improve very quickly.

### 5.4. The Impact of the Coefficient of the Green Input Cost

As the partial derivative of the optimal decisions of supply chain members regarding the green input cost coefficient is too complex, the numerical analysis was carried out, and the parameters were set as follows: $\alpha =9,w_{n}=8,w_{r}=4,c_{n}=1,\delta =1,\theta =2,\beta =0.2$. Figure 2a–f show the optimal decisions of supply chain members with the change in parameter λ.

It can be seen from Figure 2a that, with the increase in λ, the optimal recovery effort level in the four supply chain decision-making models decreases. This is because λ increases, which means an increase in green inputs, so the decision maker will choose to reduce the recycling efforts to reduce the production costs. 

It can be seen from Figure 2b,c that the greenness levels in the four supply chain decision-making models decrease with the increase in λ. In model R-M-M and model C, the impact of λ on the greenness levels of the two kinds of products is significantly greater than that in the other two models. Furthermore, in model R-M-M and model C, the optimal greenness levels are also significantly higher than those in the other two models. 

It can be seen from Figure 2d,e that the retail prices in the four supply chain decision-making models decrease with the increase in λ. In model R-M-M and model C, the impact of λ on the retail prices of the two kinds of products is significantly greater than that in the other two models. This is because, as the greenness level decreases with the increases in λ, the decision makers have to lower the retail price to attract customers. Moreover, when there is cooperation between the manufacturer and the retailer, the retailer’s decision is more likely to be impacted by the manufacturer. 

It can be seen from Figure 2f that the total profit of the supply chain decreases in the four decision-making models when λ increases. This is because, with the increases in λ increases, the green input cost increases, which inevitably damages the total profit of the supply chain.

## 6. Conclusions

In this study, we built a green, low-carbon supply chain consisting of a green manufacturer, a green remanufacturer and a retailer, and we obtained the optimal greenness level, recycling effort level, and pricing decision by implementing four different decision-making models. Then, we analyzed the impacts of the recycling cost coefficient, the green investment cost coefficient and other parameters on the optimal decisions. This study contributes to the literature by considering the optimal decisions in green, low-carbon supply chains for different types of manufacturers. This study brings together the literature on the selling channel selection strategy and the collection channel selection strategy in green supply chains. The impacts of the remanufacturing cost savings, the recovery cost coefficient and the green input cost coefficient on the optimal decision and the firms’ profits are discussed. We obtained the following results:(1)The increase in unit cost savings from remanufacturing can make a firm or alliance engaged in remanufacturing in the supply chain pay more for recycling efforts and produce more green remanufactured products. However, it will not have a positive impact on the greenness of the new products, and the increase in unit cost savings from remanufacturing will also increase total supply chain profits;(2)With the increase in the recycling cost coefficient and the green investment cost coefficient, the optimal recycling effort level and the optimal greenness level of the remanufactured product decrease, and the total profit of the supply chain decreases;(3)When there is a cooperation relationship between the manufacturer and the retailer, the optimal recycling effort level and the optimal greenness level of the new product and the remanufactured product are the highest.

In this study, we only discussed the optimal decisions in green, low-carbon supply chain pertaining to the competition and cooperation relationships between different types of manufacturers. Various interesting questions can be analyzed in the future. First, in the future, we will consider the retailer as the Stackelberg leader in a green, low-carbon supply chain and discuss the optimal decisions pertaining to the competition and cooperation relationships between different types of manufacturers. Second, we will also consider a situation in which the manufacturer can both produce the new green, low-carbon products and recycle the waste green, low-carbon product; then, a cooperative and competitive relationship can be formed between the manufacturer and the remanufacturer. All of the above questions will be explored in the near future.

## Figures and Tables

**Figure 1 ijerph-19-15111-f001:**
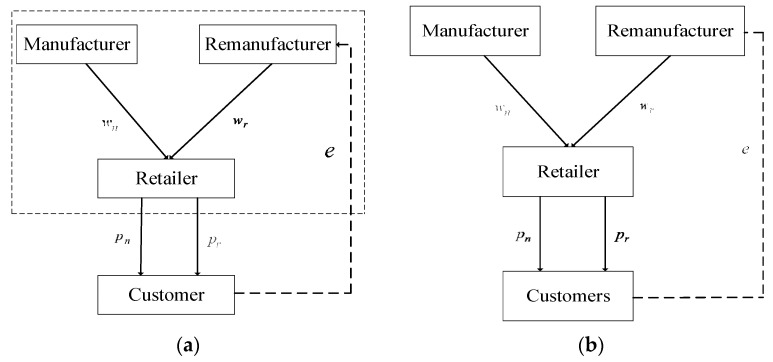
Four different green supply chain theoretical models. (**a**) model C; (**b**) model D; (**c**) model M-M; (**d**) model R-M-M.

**Figure 2 ijerph-19-15111-f002:**
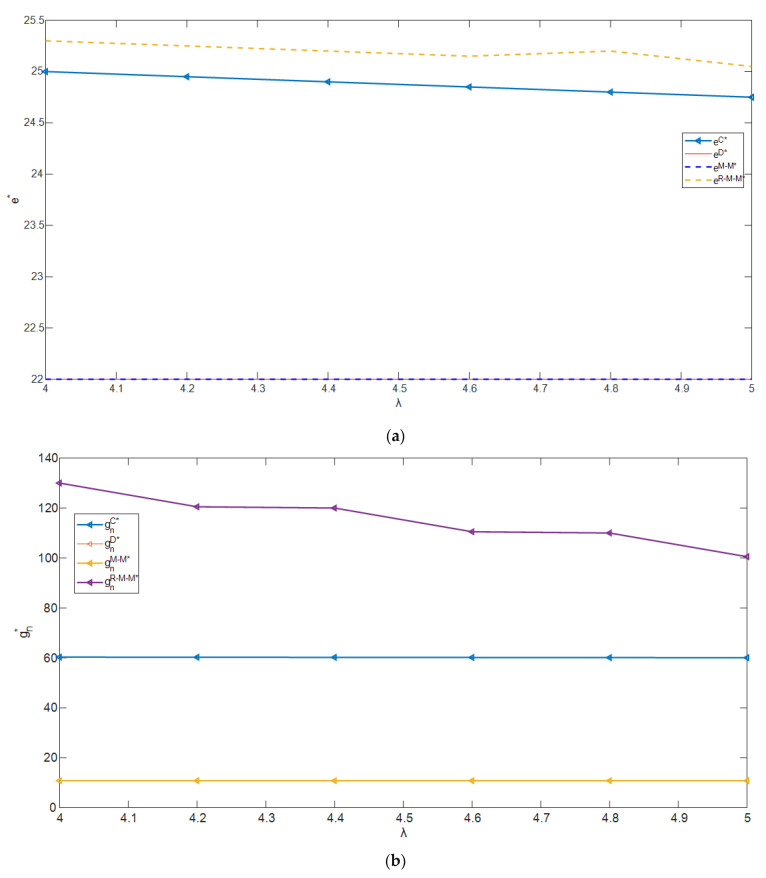
The impact of λ on the equilibrium results. (**a**) The impact of λ on $e^{*}$; (**b**) the impact of λ on $g_{n}^{*}$; (**c**) the impact of λ on $g_{r}^{*}$; (**d**) the impact of λ on $p_{n}^{*}$; (**e**) the impact of λ on $p_{r}^{*}$; (**f**) the impact of λ on $\pi^{*}$.

**Table 1 ijerph-19-15111-t001:** The difference between our paper and the previous research.

Literature	Forward Supply Chain	Reverse Supply Chain	Carbon Emission Constraint	The Competition and Cooperation Relationship
	Pricing Decision	Greenness Level Decision		
Swami and Shah [1]	√			
Chen [2]	√			
Wang et al. [4]	√			
Zhang and Li [5]	√			
Yan et al. [7]	√			
Qu et al. [9]	√			
Gong et al. [10]	√			
Ofek et al. [12]	√		√	
Guide [16]		√		
Peral [19]		√	√	
Giovanni [21]		√		√
Yi et al. [25]		√	√	
Polat et al. [26]		√		
Our paper	√	√	√	√

**Table 2 ijerph-19-15111-t002:** Description of the relevant notation.

Variables	Notation	Description
Decision variables	e	The level of recycling effort of the green remanufacturer
	$g_{n}$	The greenness level of the new products
	$g_{r}$	The greenness level of the remanufactured products
	$p_{n}$	The retail price of the new products
	$p_{r}$	The retail price of the remaufactured products
Relevant parameters	δ	The unit cost savings in remanufacturing
	θ	The coefficient of the recycling cost
	λ	The coefficient of the green input cost
	$c_{n}$	The unit production cost of the new green products
	$c_{r}$	The unit production cost of the new remanufactured green products
	$w_{n}$	The wholesale price of the new green products
	$w_{r}$	The wholesale price of the new remanufactured green products
	$\alpha_{n}$	The potential market size for the new products
	$\alpha_{r}$	The potential market size for the remanufactured products
	f	The carbon emission per unit product of a new or remanufactured product
	$f_{0}$	The upper limit of carbon emissions for new or remanufactured products
	η	The coefficient of the carbon emission abatement
	τ	The sensitivity coefficient of carbon emissions per unit product of a new or remanufactured product
	$\pi^{C}$	The profit function in model C
	$\pi_{Mn}^{D}$	The manufacturer’s profit in model D
	$\pi_{Mr}^{D}$	The remanufacturer’s profit in model D
	$\pi_{R}^{D}$	The retailer’s profit in model D
	$\pi_{M}^{M-M}$	The profit of the alliance between the manufacturer and the remanufacturer in model M-M
	$\pi_{R}^{M-M}$	The retailer’s profit in model M-M
	$\pi_{RMn}^{R-M-M}$	The profit of the alliance between the manufacturer and the retailer in model R-M-M
	$\pi_{RMr}^{R-M-M}$	The profit of the alliance between the remanufacturer and the retailer in model R-M-M

Please note that Table 2 refers to parameters (i.e., the cost $c_{n}$ and $c_{r}$ and the price $p_{n}$ and $p_{r}$) in units of yuan.

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
