# Peer review of "Optimal Decisions in Green, Low-Carbon Supply Chain Considering the Competition and Cooperation Relationships between Different Types of Manufacturers"

_ijerph, 2022, doi:10.3390/ijerph192215111_

Round 1

Reviewer 1 Report

Please revise the manuscript follow the suggestions as shown in attach file. 

Author Response

Dear reviewer:

Thank you for your comments concerning our manuscript entitled “Optimal Decision of Green Low Carbon Supply Chain Considering the Competition and Cooperation Relationship Between Different Types of Manufacturers” (ID:ijerph-1999795). Those comments are all valuable and very helpful for revising and improving our paper, as well as the important guiding significance to our research. We have studied the comments carefully and have made corrections which we hope meet with approval. Revised portions are marked in yellow on the paper. The main corrections in the paper and the responses to the reviewer’s comments are attached.

Reviewer 2 Report

The paper is well-written, but there is a room to improve the English wordings.

The conclusion can be further enhanced to discuss any further research that could be carried out based on the result of this research project.

Author Response

(The authors gave the same response as above.)

Reviewer 3 Report

Overall, this paper sounds very interesting. It concerns a green low carbon supply chain. The entire idea is well-addressed. However, there is a number of considerations that might need to be improved further.

- Abstract does not highlight the research methodology used and what methods were used in data collection.

-Section 1 should be added with relevant references. The existing seems to ignore the previous studies in regard to the topic. Section 1 seems too long, but there is lacking references to support statements.

-The author might need to consider adding a matrix table to support the idea of the author in the literature review section. Having the table will be helpful to the reader to understand what is the main gap of the study.

-The development of the models has to be addressed the process behind it. What are the processes involved in the framework development? Was any preliminary discussion done? if yes please explain it.

-The results presented based on different types of models have to be tested and validated. But the author seems to overlook it.

-The conclusion should address the limitation and contribution to the body of knowledge and practice. The author has to add the information distinctively.

Author Response

(The authors gave the same response as above.)

Round 2

Reviewer 1 Report

This manuscript can be publish.